# Teaching in Times of the COVID-19 Pandemic: A Pilot Study on Teachers’ Self-Esteem and Self-Efficacy in an Italian Sample

**DOI:** 10.3390/ijerph18158211

**Published:** 2021-08-03

**Authors:** Stefania Cataudella, Stefano Mariano Carta, Maria Lidia Mascia, Carmelo Masala, Donatella Rita Petretto, Mirian Agus, Maria Pietronilla Penna

**Affiliations:** Department of Pedagogy, Psychology, Philosophy, Faculty of Humanistic Studies, University of Cagliari, 09123 Cagliari, Italy; cartast@unica.it (S.M.C.); marialidia.mascia@unica.it (M.L.M.); masalacarmelo@gmail.com (C.M.); drpetretto@unica.it (D.R.P.); mirian.agus@unica.it (M.A.); penna@unica.it (M.P.P.)

**Keywords:** teaching, self-esteem, self-efficacy, learning disabilities, distance learning

## Abstract

The aim of this research was to investigate the impact of the COVID-19 pandemic on teachers, particularly on their self-esteem and self-efficacy, their difficulty in the transition to distance learning, the difficulty of students, and specially of students with learning disabilities (LDs students), as perceived by teachers. 226 teachers were invited to complete an online questionnaire. Our results showed lower self-esteem and lower self-efficacy by the teachers compared with the normative sample. Self-esteem and self-efficacy also decrease in teachers with greater service seniority at work. Teachers perceived a greater difficulty in students than in their own difficulty. The concentration of the school system’s efforts on the massive and, for long periods, exclusive organisation of distance learning risks favouring only cognitive aspects to the detriment of affective dynamics. This aspect could make teaching more complex for teachers and learning poorer for students, impoverishing the complex relational process that forms the basis of the learning process.

## 1. Introduction

A body of literature showed the fundamental role played by the socioemotional aspects in the school context [1]. When the school is also interested in promoting student’s socioemotional learning, apart from traditional outcomes, many educational benefits, including students’ academic achievements, affects, behaviours, and motivation, are reaped [2,3,4].

A lot of research on socioemotional issue shows the important role played by a positive quality of teacher–student relationships in influencing learning both directly and indirectly [5,6,7,8,9,10].

Some studies have contributed to the understanding of student–teacher dynamics and how this relationship in the school context results in student learning. Several interpersonal variables were identified from the literature as positively related to learning: immediacy [11,12] communicator style [13,14] affinity-seeking [15], self-disclosure [16], humour [17], job satisfaction [18], enthusiasm [19], self-esteem [20], self-efficacy [21], etc.

The importance of relationships, and in particular those in school settings, is a theme that has begun to appear in the forefront since the past year, as policy-makers and educators realise the importance of human socioemotional aspects in the relationship between teachers and students amidst the sudden shift from face-to-face to online schooling during the ongoing COVID-19 pandemic.

Teachers suddenly have to deliver their lessons using technological tools, including through specific online platforms, in order to reach out to the students. Students have been deprived of social face-to-face interaction among peers, and teachers and parents began to be more involved because of the need for monitoring school lessons at home.

Although some teachers were ready to face the situation, a large majority had to adapt their teaching in a short time without training, with insufficient capacity, and little preparation. This unexpected and rapid transition to online learning has led to a multiplication of teachers’ strategies for distance learning in lectures, tutorials, project groups, lab works, and assessments [22].

Several studies showed that the teaching profession has experienced many periods of crisis and discontent throughout history. The profession has been strongly guided by high rates of professional dissatisfaction, stress linked to the vastness of the bureaucracy, excessive workload imposed on teachers, incidents of indiscipline, precariousness and professional instability, teachers’ lack of motivation and/or interest, and the growing use of technology in teaching [23,24,25].

The current pandemic has given rise to a crisis that particularly involves the last aspect, which is the impact of digital technology on teachers’ school practice. COVID-19 has once again highlighted the problematic relationship between teaching and new technologies, both in the international than in the national context [26,27].

For example, a study [28] on the professional well-being of Portuguese teachers showed that the pandemic has led to a shift in teachers’ perception from fair positivity to concern about their professional future. Additionally, in the Italian context, this aspect of increasing technology use is emphasised on its influence on job satisfaction [26,27].

Job satisfaction involves psychological, physiological, and environmental conditions and factors that, together, guarantee positive feelings towards work [29], which, in turn, increase the rate of productivity and sense of well-being. Among the variables found in the literature, self-esteem and self-efficacy were found to play an important role in job satisfaction and in the ability to meet or address changes. The influential variables on teachers’ job satisfaction affect teacher–student and teacher–parent communication as well as the aspect of collaboration.

Rosenberg [30] describes self-esteem as individuals’ positive and negative self-perception. Self-esteem is an individual’s consideration of his/her own self as competent and important, as well as perceiving oneself as successful and valuable [31]. Several studies showed that employees with high-level of self-esteem had high-level of job satisfaction [32,33]. Mbuva [20] stated that teachers’ self-esteem is important for their success in teaching and that teachers’ positive and high esteem, in turn, positively affects students’ self-esteem and learning processes.

Another variable that was found to positively influence job satisfaction, was self-efficacy [34]. Self-efficacy is a person’s conviction in their ability to succeed in a particular situation [35]. Perceptions of teacher self-efficacy correspond to judgements about the teacher’s personal ability to achieve the desired results in terms of student engagement and learning. The concept of self-efficacy derives from Bandura’s social-cognitive theory of behavioural change, It refers to a teacher’s belief in his/her ability to successfully cope with tasks, obligations and challenges related to his/her professional role [36]. Teacher self-efficacy is also defined as teacher’s perception in motivating students to learn [37]. Another definition describes the teachers’ belief that they can influence their students’ learning process. The literature has shown that teachers with high levels of self-efficacy experience higher levels of job satisfaction and lower levels of job-related stress [38]. Teacher self-efficacy was positively related to class-average achievement and interaction quality [21].

Several studies on distance education showed that despite teachers’ limited experience in this type of education in terms of technical skills, time management, knowledge and attitude in online education, teachers can cope with the trends in distance learning [39]. Some studies highlighted the relationship difficulties frequently experienced by regular students (lack of interaction and feedback, difficulty to start and maintain communication, ambiguity of posted messages, technical problems disrupting conversations, etc.) [40,41,42] and by students with learning disabilities (LDs) (high levels of social distress and loneliness) [43,44,45].

Distance learning is a psychological process supported by e-technology, but learning is a social activity. The use of distance learning can improve the learning of students and students with LDs only where a supportive context is present [45].

The success of distance learning also depends on education management and the quality of the students’ home learning environment. Because this pandemic is still on-going, it is essential to determine how the teachers, who are the main facilitators of education, are adjusting to this transition.

At this stage, it is interesting to investigate teachers’ perceptions in relation to this sudden transition from face-to-face teaching to distance teaching; in particular, we are interested in investigating their perceptions of difficulties (both for themselves and for their students) and their self-assessments in terms of self-esteem and self-efficacy.

## 2. Materials and Methods

The aim of this study is to investigate the impact of the COVID-19 pandemic on teachers, particularly regarding their perspectives on the following:The effect of changes because of pandemic-related changes on self-esteem and self-efficacy through comparison with the normative samples;Their level of difficulty in the transition to distance learning;The level of difficulty of students, particularly students with LDs, as perceived by teachers in the transition to distance learning.

We also aimed to investigate the effect of the variables “school type” and “level of service” (low <14 years vs. high >14 years) on the perception of difficulties and on self-esteem and self-efficacy.

### 2.1. Participants

Our participants were recruited through non-probability sampling across 28 schools in some areas of Sardinia (Italy), which produced a total of 226 respondents (Figure 1 depicts phases of recruiting): 199 females (88.1%) and 27 (11.9%) male teachers. Of the total, 23 (10.2%) participants were aged <35 years, 93 (41.2%) were between 36–45 years, 66 (29.2%) were between 46–55 years and 44 (19.5%) were age >56 years. Twenty-eight participants (12.4%) taught in kindergarten school, 80 (35.4%) participants in primary school, 49 (21.7%) in junior high school, and 69 (30.5%) in high school. A total of 143 (63.3%) participants had professional stability, while 82 (36.3%) participants had professional instability (one teacher did not give this data). Only 44.7% of teachers have received professional training on learning disability. The descriptive statistics were showed in Table 1.

The study was approved by the Ethics Committee of the University of Cagliari, Italy (Prot. n. 0073815).

### 2.2. Instruments

All teachers were invited via e-mail to complete an online questionnaire using Google Forms as a platform. A work session lasting about 25 minutes was carried out, organised in different sections.

The first section of the questionnaire assessed the demographic variables (age, gender, level of education, years of seniority at work, type of contract, experience in LDs field, etc.).

In order to identify specific situations and perceptions during this period of the COVID-19 pandemic, the second section included a number of relevant questions: (Q15) “During the COVID-19 emergency, did your school set up online teaching activities? (Q16) If yes, was an official platform used? (Q17) If yes, which one? Gsuite, Weschool, Adobe connect, Microsoft Teams, No platform, only e-mail, Other”; (Q18) “What is the level of difficulty you encountered in using the platform?”, (Q19) “What is the level of difficulty encountered by your students in using the platform?”, (Q20) “What is the level of difficulty encountered by your students (with LDs) in using the platform?”, (Q21) “What is the level of difficulty encountered by your students (with SEN) in using the platform?”, (Q22) “What is the level of difficulty encountered by your students (with H) in using the platform?”, (Q23) “How was the level of cooperation from parents?”, (Q24) “How was the level of student’s participation?”.

The second section of the questionnaire assessed the following variables:

#### 2.2.1. Self-Esteem

The Rosenberg Self-Esteem Scale is a self-report questionnaire, introduced in 1965 by Rosenberg. It comprises five positive and five negative worded items (for the analyses, the five negative items were reversed). In this study, we use the Italian version by Prezza, Trombaccia and Armento [46]. The rating scale was a four-point Likert-type scale, ranging from 1 (strongly disagree) to 4 (strongly agree). Its internal consistency, analysed by the Cronbach coefficient, was 0.84.

#### 2.2.2. Self-Efficacy

The Teacher Self-Efficacy Scale [37] was administered in its Italian adaptation by Biasi and Dominici [47]. The scale comprises 24 items organised in three factors: «Efficacy for Student Engagement» (8 items), «Efficacy for Instructional Strategies» (8 items), and «Efficacy for Classroom Management» (8 items). Regarding the scales assessed, the “teachers’ perception of self-efficacy” relates to their ability to gain students’ commitment, the ability to choose appropriate teaching strategies, and their ability to manage the classroom.

The “Efficacy for Student Engagement” sub-scale measures teachers’ sense of efficacy in motivating the students. A teacher who is confident of being able to motivate the student, would need to be involved and committed in the study in order to affect the results.

The “Efficacy for Instructional Strategies” sub-scale identifies the teacher perceptions in using appropriate teaching strategies.

The “Efficacy for Classroom Management” sub-scale evaluate the teacher perceptions in managing the class in a functional way.

Cronbach’s alpha in the Italian sample was good (0.97).

### 2.3. Data Analysis

With the aim of eliciting answers to research questions, an exploratory and descriptive approach was applied [48]. The descriptive analyses were applied in relation to all the investigated variables. Thereafter the non-parametric and parametric inferential analyses were conducted.

One sample *t* test was carried out to compare the mean scores of the participants in relation to the normative samples. The bivariate correlations (Pearson’s r coefficient) evaluated the relationships between continuous variables. The multivariate analysis of variance (MANOVA) was applied to assess the differences in relation to the level of seniority (low and high seniority—evaluated by the median split) and the type of school (pre-school, primary school, lower secondary school, upper secondary school), regarding the dimensions of self-efficacy and self-esteem. The non-parametric Friedman test and Mann–Whitney test were carried out to explore the differences in ordinal variables regarding the level of seniority and type of school.

The two-step cluster analysis was applied in order to identify the participants’ groups regarding dimensions. This statistical technique applied two phases of clustering [49]: the first pointing to grouping cases into pre-clusters, and the second to allocating all cases in clusters. The first phase aims to reduce the size of the matrix that contains distances between all possible pairs of cases. This clustering, conducted on continuous variables, makes use of the Euclidian algorithm. For this study, the individuals were the objects to be clustered, while the variables characterised attributes on which clustering was grounded. For the second step, the pre-clusters were grouped using the hierarchical clustering algorithm. Proposing a range of clustering solutions, this step also involved condensing to the best number of clusters based on the Schwarz’s Bayesian information criterion (BIC). After the identification of the cluster solution, t tests for continuous variables were carried out to examine the importance of each variable in a cluster [50,51].

The data analyses were carried out using the opensource software Jamovi (release—1.6.15; jamovi project 2021, https://www.jamovi.org (accessed on 5 June 2021)) [51] and SPSS (release 24—IBM Corporation, Armonk, NY, USA) [52].

## 3. Results

### 3.1. Comparisons with Normative Samples for Self-Esteem and Self-Efficacy Scores

In order to compare the mean scores of the participants to the scores obtained by the normative samples in the validated instruments that were applied, the one sample *t* test was carried out (Table 2). The data highlighted noteworthy differences regarding the aspect of self-esteem, with the participants showing significantly lower scores compared to the normative sample (*t* = −32.700; df = 225; *p* < 0.001; Cohen’s d = −2.170). A significant difference was found regarding the dimension of self-efficacy for instructional strategies, consistently revealing significantly lower scores for the teachers taking part in our research (*t* = −2.020; df = 225; *p* = 0.044; Cohen’s d = −0.134) compared with the normative sample. For the dimensions assessing self-efficacy for classroom management and self-efficacy for student engagement, no significant differences were highlighted between normative sample and participants.

### 3.2. Bivariate Correlations and MANOVA Regarding Type of School and Level of Seniority for Self-Esteem and Self-Efficacy Scores

In order to evaluate the linear relations between the inquired dimensions, the Pearson’s r coefficient was applied (Table 3). The data were found to emphasise the significant positive correlations between self-esteem and self-efficacy and the negative correlations between years of seniority and self-efficacy.

A MANOVA with two factors (2 × 4) was applied to evaluate the differences in relation to the level of seniority (low and high seniority, appraised by the median split. Low: less than 14 years’ seniority; high: over 15 years’ seniority) and the type of school (pre-school, primary school, lower secondary school and upper secondary school), regarding the dimensions of self-efficacy and self-esteem. The use of Wilk’s Lambda multivariate test highlighted a significant principal effect for the level of seniority (Wilks’ Lambda = 0.948; F = 2.942; df = 4; 214; *p* = 0.021). There were no other significant effects, found in relation to the principal effect of the type of school (Wilks’ Lambda = 0.913; F = 1.647; df = 12; 566; *p* = 0.075) and to the interaction between the level of seniority and type of school (Wilks’ Lambda= 0.972; F = 0.506; df = 12; 566; *p* = 0.911). Significant effects regarding the level of seniority were observed regarding all assessed dimensions of self-esteem and self-efficacy in teaching (see Table 4 and Table 5). Table 5 reports the descriptive values (mean, standard deviation) relating to the significant effect of the level of seniority factor, highlighting that teachers with lower level of seniority showed higher scores in all scales (self-esteem and self-efficacy).

### 3.3. Non-Parametric Analyses in Order to Evaluate Differences Regarding the Ordinal Variables

A specific section of the questionnaire assessed the level of difficulty perceived by teachers in using digital platforms.

In order to evaluate if the level of difficulties perceived by the teacher was different in relation to their perceptions of different actors in school (oneself, students, LDs students—Questions Q18, Q19, Q20) Friedman’s non-parametric test for repeated measures at ordinal level was applied (Table 6). In this evaluation, we excluded teachers working in pre-school, as students under the age of 6 tended not to use the platforms for structured learning activities. Subjecting the data to the Freedman test revealed a significant effect (Chi Squared = 125.00; df = 2; *p* < 0.001). Pairwise comparisons with the Durbin–Conover Test emphasised significant differences between all couples of questions. Teachers’ perceptions of their personal difficulties were significantly lower than their perceptions of students’ difficulties, and specifically of LD students.

The Mann–Whitney test for repeated measures was applied for the questions assessing the level of participation of student and parents (Q23, Q24). This assessment comparison emphasised a higher level of participation of the students than their parents (Wilcoxon = 1294, *p* < 0.001).

Thereafter, we evaluated the level of difficulties perceived in relation to the level of seniority (low, high). The Mann–Whitney test highlighted a significant difference regarding the level of difficulty experienced by teachers (Mann–Whitney U = 5010; *p* = 0.008; Effect size = 0.194) that was higher for persons with higher seniority.

We compared the difficulty perceived by the teachers in relation to the type of school. The Kruskal–Wallis test highlighted significant differences only regarding the level of collaboration of parents (KW = 36.019; df = 2; *p* < 0.00; Effect size = 0.209), i.e., that significant differences were found to be higher in the primary school than in higher types of schools.

### 3.4. Two Steps Cluster Analysis

Finally, two-step cluster analysis was carried out to identify similar groups or “clusters” of people within this study’s data sets [53,54]. This technique was applied on the continuous scale scores in the psychological dimensions of self-esteem and self-efficacy achieved by the teachers. A solution characterised by three clusters was identified (Schwarz Bayesian Criterion (BIC) = 1330.654; modification BIC = −189.941; ratio of modifications BIC = 0.710; ratio of distance measures = 1.736). This was also confirmed following a random subdivision of the total sample into two subgroups. Three clusters of teachers were identified: the first comprised of 47 participants (21.5%), the second by 104 participants (47.5%), and the third by 68 (31.1%) (Table 7).

As seen in Table 7, cluster 1 is characterised by teachers with a seniority of service >14 years, having a predominantly open-ended contract (68.1%) and lower levels of self-esteem and self-efficacy than in the average of the sample participants. Cluster 2 is also characterised by teachers with a seniority of service >14 years, with an indefinite duration but with levels of self-esteem and self-efficacy higher than in the average of the sample and lower than the average of cluster 3, which is characterised instead by teachers with a seniority of service <14 years and with a type of contract that was mainly fixed-term. These data suggest that a relevant difference between Clusters 1 and 2 could be recognised in the training of teachers belonging to Cluster 2, in terms of the use of technology prior to the ongoing pandemic; this aspect might have a relevant role supporting a more comfortable transition to distance learning. Furthermore, this aspect might be supported by the features of Cluster 3: indeed, the fewer difficulties encountered by younger teachers might could be attributable to their better knowledge of technology, regardless of their training.

## 4. Discussion

The aim of this pilot study was to investigate the impact of the COVID-19 pandemic on teachers. In particular, we investigated teachers’ perspectives on the following: the impact of pandemic-related changes on self-esteem and self-efficacy through comparison with the normative sample; their level of difficulty in the transition to distance learning; and the level of difficulty of students, and specially of students with LDs, as perceived by teachers. We also investigated the effect of the variables ‘school type’ and ‘level of service’ (low <14 years vs. high >14 years) on the perception of difficulties and on self-esteem and self-efficacy.

Our results allow some important reflections on the pressures to which schools were subjected to during this period of the COVID-19 dilemma. The observation point we have used in this first phase of the study is from the perspective of the teachers. Political decisions, aimed at managing the pandemic, forced Italian schools to reorganise their teaching homogeneously without considering the differences in the national territory. The required transition from face-to-face teaching to distance teaching in a sudden and discontinuous way has brought out the gap on the national territory both on teachers’ training and on the concrete possibility for families to support this transition to online platforms [55].

In consideration of our aim, the data highlighted differences regarding the aspects of self-esteem and self-efficacy for instructional strategies wherein we found that participants significantly scored lower than the normative sample. As for the dimensions assessing self-efficacy for classroom management and self-efficacy for student engagement, no significant differences were found between the normative samples and the participants. The use of the Wilk’s Lambda multivariate test highlighted that the teachers’ perceptions of their personal difficulties were significantly lower compared to their perceptions of students’ difficulties, particularly LDs students, while the Mann–Whitney test highlighted a significant difference regarding the level of difficulty experienced by teachers having greater seniority.

A first consideration underscores a stronger focus on self-esteem and self-efficacy for instructional strategies vs. what we may call the ‘relational/interactional’ dimensions of efficacy for classroom management and self-efficacy for student engagement. This may indicate that teachers perceive their personal difficulties more as belonging to ‘what they are’ than to what they do, or may do. This may be explained by the teachers’ lack of a clear range of strategies for interventions and management, which in turn may be explained by this never-before experienced situation of long-term online distance teaching. It seems that the teachers, lacking a clear representation of possible alternative teaching and relational strategies, failed to assess the potential solutions to this challenging teaching environment [26,27,56,57,58]. Moreover, the data on scores on these self-related dimensions (self-esteem and self-efficacy for instructional strategies) points to a possible negative development, namely the failure to form positive and constructive coping responses and the inability to find possible active solutions to frustrating tasks or predicaments.

A second consideration that seems to emerge from our data is that the finding of difficulties as increasing with seniority may indicate that, amidst a challenging new predicament, seniority, when not accompanied by proper training on interactional/relational dimensions, may be more of an obstacle than a resource. As experience brings with itself a tendency to learn and therefore repeat what has been learned, in this case, before developing new teaching strategies, the senior teachers might need to partially reconsider what they have learned throughout their career—a psychological and professional challenge that the junior teachers simply do not go through.

Teachers with greater seniority also tend to focus more on students and students with LDs, i.e., teachers tend to perceive a greater difficulty in students than inthemselves. In general, while the great pressure deriving from the particular predicament we are analysing probably directed the focus on technology and not on the relationship with their students, teachers with greater seniority mainlyfocussed on the relational components as, due to their experience, they inevitably have greater perceptions of the difficulty of others, particularly that of students.

## 5. Conclusions

The concentration of the school system’s efforts on the massive and, for a long time, exclusive organisation of distance learning risks favouring only cognitive aspects to the detriment of affective dynamics. This aspect could make teaching more complex for teachers and effect poorer learning in students. This raises a further reflection on the importance of social relationships. Even though distance learning allows virtual meeting for sharing time and didactic contents, it takes away many characteristics of the didactic experience that are fundamental in social relationships, especially in subjects of a developmental age [59]. The absence of relational experience space, with its verbal and non-verbal communicative characteristics, impoverishes the complex relational process that forms the basis of the learning process [60].

The results of this study allow us in part to confirm some data in the literature: data that identify in the relational variables those that can trigger a virtuous process that increases the self-esteem and self-efficacy of teachers who can thus involve students in the learning process, both affectively and cognitively, resulting in academic success.

We believe that our study results may be useful for planning possible teacher’s training based on emotional support related to the self-related dimensions, on relational/interactional field and, finally, on a specific preparation aimed at giving prerequisite necessary to reorganise the learning processes through new relational and teaching skills and strategies.

Our work has several limitations, including a small sample that needs to be extended (numbers of subjects and representative of school orders), from the perspective of teachers rather than that of students and families. We intend to continue our work in these directions.

It would be interesting in future research, also from a longitudinal perspective, to analyse the effects and changes in the relational processes underlying learning.

## Figures and Tables

**Figure 1 ijerph-18-08211-f001:**
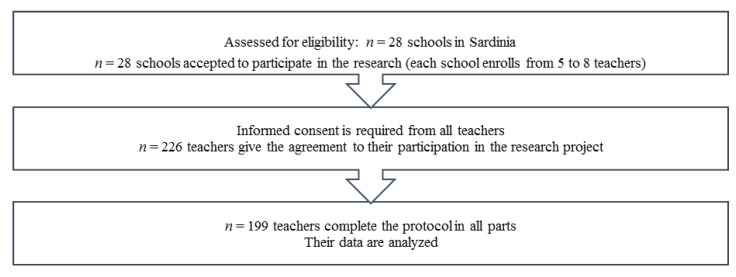
Recruiting phases.

**Table 1 ijerph-18-08211-t001:** Descriptive statistics of the assessed variables.

	Overall (*n* = 226)
Age	
between 36–45 years	93 (41.2%)
>56 years	44 (19.5%)
between 46–55 years	66 (29.2%)
<35 years	23 (10.2%)
Gender	
females	199 (88.1%)
males	27 (11.9%)
Level of School	
university degree	112 (49.6%)
high school diploma	55 (24.3%)
university diploma	7 (3.1%)
post-graduate training	52 (23.0%)
Type of School You Teach In	
kindergarten school	28 (12.4%)
primary school	80 (35.4%)
junior high school	49 (21.7%)
high school	69 (30.5%)
Years of Seniority at Work	
*n*-miss	1
mean (sd)	14.7 (11.2)
range	0.0–42.0
Levels of Years of Seniority at Work	
*n*-miss	1
<14years	127 (56.4%)
>15 years	98 (43.6%)
Specialisation in Student Support	
no	186 (82.3%)
yes	40 (17.7%)
Learning Disabilities and/or Disabilities Referent	
no	215 (95.1%)
yes	11 (4.9%)
Hours of Work	
*n*-miss	5
timeslot	9 (4.1%)
part time	24 (10.9%)
full time	188 (85.1%)
Professional Training on Learning Disability	
no	125 (55.3%)
yes	101 (44.7%)
Type of Contract	
*n*-miss	1
limited time	82 (36.4%)
undefined time	143 (63.6%)
During the COVID-19 emergency, did your school set up online teaching activities?	
yes	218 (96%)
If yes, was an official platform used?	
yes	186 (82%)
Self-Esteem	
mean (sd)	23.5 (3.3)
range	10.0–32.0
Self-Efficacy for Student Engagement	
mean (sd)	6.8 (2.1)
range	1.0–9.0
Self-Efficacy for Classroom Management	
mean (sd)	6.6 (2.1)
range	1.0–9.0
Self-Efficacy for Instructional Strategies	
mean (sd)	6.9 (2.1)
range	1.0–9.0

**Table 2 ijerph-18-08211-t002:** Comparison between mean scores of participants and normative samples.

	Self Esteem	Self-Efficacy for Instructional Strategies	Self-Efficacy for Classroom Management	Self-Efficacy for Student Engagement
Meannormative sample	30.61	7.14	6.8	6.98
Mean (sd) participants	23.50 (3.26)	6.86 (2.10)	6.580 (2.07)	7.50 (2.08)
Median participants	24.00	7.63	7.250	7.50
Standard Error participants	0.217	0.140	0.138	0.139
Student’s *t* (df)	−32.700 (225)	−2.02 (225)	−1.63 (225)	−1.29 (225)
*p*	<0.001 ***	0.044 *	0.105	0.200
Mean difference (95% CI)	−7.100 (−7.530; −6.670)	−0.282 (−0.557; −0.008)	−0.224 (−0.495; 0.047)	−0.177 (−0.448; 0.094)
Cohen’s d effect size (95% CI)	−2.170 (−2.410; −1.930)	-0.134 (−0.265; −0.003)	-0.108 (−0.239; 0.022)	-0.085 (−0.216; 0.045)

Note: * *p* < 0.05; *** *p* < 0.001.

**Table 3 ijerph-18-08211-t003:** Linear correlations between years of seniority, self-esteem and self-efficacy scores.

		Self-Esteem	Self-Efficacy	Self-Efficacy	Self-Efficacy
			for Student	for Classroom	for Instructional
			Engagement	Management	Strategies
Self-esteem	r	—			
	*p*	—			
Self-efficacy for student engagement	r	0.376 ***	—		
	*p*	<0.001	—		
Self-efficacy for classroom management	r	0.354 ***	0.964 ***	—	
	*p*	<0.001	<0.001	—	
Self-efficacy for instructional strategies	r	0.365 ***	0.962 ***	0.952 ***	—
	*p*	<0.001	<0.001	<0.001	—
Years of seniority at work	r	−0.117	−0.147 *	−0.139 *	−0.168 *
	*p*	0.079	0.028	0.038	0.012

Note: * *p* < 0.05; *** *p* < 0.001.

**Table 4 ijerph-18-08211-t004:** Manova—Univariate Tests.

	Dependent Variable	Sum of Squares	Df(b;w)	Mean Square	F	*p*	Effect Size Partial Eta Squared
Level of school	Self-esteem	61.03	3;217	20.34	1.969	0.120	0.031
	Self-efficacy for student engagement	19.70	3;217	6.57	1.578	0.196	0.017
	Self-efficacy for classroom management	23.73	3;217	7.91	1.905	0.130	0.017
	Self-efficacy for instructional strategies	20.69	3;217	6.90	1.617	0.186	0.018
Level of seniority	Self-esteem	74.32	1;217	74.32	7.193	0.008 **	0.024
	Self-efficacy for student engagement	30.89	1;217	30.89	7.426	0.007 **	0.020
	Self-efficacy for classroom management	26.95	1;217	26.95	6.490	0.012 *	0.021
	Self-efficacy for instructional strategies	35.28	1;217	35.28	8.269	0.004 **	0.022
Level of school* level of seniority	Self-esteem	18.62	3;217	6.21	0.601	0.615	0.008
	Self-efficacy for student engagement	8.45	3;217	2.82	0.677	0.567	0.009
	Self-efficacy for classroom management	9.32	3;217	3.11	0.748	0.525	0.010
	Self-efficacy for instructional strategies	7.56	3;217	2.52	0.591	0.622	0.008

Note: * *p* < 0.05; ** *p* < 0.01.

**Table 5 ijerph-18-08211-t005:** Means and standard deviations regarding the levels of seniority.

Level of Seniority at Work		Self-Esteem	Self-Efficacy for Student Engagement	Self-Efficacy for Classroom Management	Self-Efficacy for Instructional Strategies
<14 y *n* = 127	mean	23.976	7.114	6.867	7.187
	sd	2.958	1.973	1.962	1.981
>15 y *n* = 98	mean	22.887	6.387	6.183	6.419
	sd	3.557	2.134	2.153	2.182
Total	mean	23.502	6.797	6.569	6.852
	sd	3.2707	2.072	2.073	2.101

**Table 6 ijerph-18-08211-t006:** Level of difficulty perceived by teachers in using digital platforms.

Options of Response	Q18	Q19	Q20	Q23	Q24
none—fr(fr%)	56 (25%)	10 (4.4%)	7 (3.1%)	13 (5.8%)	2 (0.9%)
low—fr(fr%)	96 (42%)	70 (31%)	40 (18%)	43 (19%)	33 (15%)
medium—fr(fr%)	61 (27%)	98 (43%)	84 (37%)	111 (49%)	122 (54%)
high—fr(fr%)	13 (5.8%)	48 (21%)	38 (17%)	59 (26%)	69 (31%)
missing—fr(fr%)	0	0	57 (25%)	0	0
median	1	2	2	2	2
Total sample mean	1.14	1.81	1.91	1.97	2.19
sd	0.855	0.772	0.758	0.830	0.649
MDN	1.00	2.00	2.00	2.00	2.00
Low level of seniority at work (<14 years) mean	1.02	1.78	1.85	1.94	2.16
sd	0.828	0.737	0.791	0.826	0.641
MDN	1.00	2.00	2.00	2.00	2.00
High level of seniority at work (>14 years) mean	1.28	1.84	1.99	2.01	2.24
sd	0.870	0.815	0.712	0.837	0.661
MDN	1.00	2.00	2.00	2.00	2.00
Primary school mean	1.24	1.94	2.02	2.33	2.29
sd	0.945	0.832	0.680	0.689	0.697
MDN	1.00	2.00	2.00	2.00	2.00
Junior high school mean	1.04	1.88	1.89	2.14	2.24
sd	0.789	0.634	0.859	0.677	0.560
MDN	1.00	2.00	2.00	2.00	2.00
High school mean	1.10	1.61	1.83	1.43	2.03
sd	0.789	0.752	0.740	0.813	0.641
MDN	1.00	2.00	2.00	2.00	2.00

Note: (Q18) “What is the level of difficulty you encountered in using the platform?”, (Q19) “What is the level of difficulty encountered by your students in using the platform?”, (Q20) “What is the level of difficulty encountered by your students with LDs in using the platform?”, (Q23) “How was the level of cooperation from parents?”, (Q24) “How was the level of student’s participation?”.

**Table 7 ijerph-18-08211-t007:** Description of the three clusters according to the variables assessed.

Center of Clusters	Cluster				
	1 (*n* = 47)	2 (*n* = 104)	3 (*n* = 68)	Combined (*n* = 219)	F(df) *p*
Self-esteem mean (SD)	21.787 (4.422)	23.615 (3.021)	24.647 (2.057)	23.543 (3.284)	11.609 (2) *p* < 0.001
Self-efficacy for student engagement mean (SD)	3.343 (1.116)	7.713 (0.844)	7.830 (0.966)	6.812 (2.047)	402.684 (2) *p* < 0.001
Self-efficacy classroom management mean (SD)	3.263 (1.120)	7.525 (0.969)	7.446 (1.200)	6.586 (2.045)	284.141 (2) *p* < 0.001
Self-efficacy for instructional strategies mean (SD)	3.391 (1.188)	7.752 (0.911)	7.906 (0.956)	6.864 (2.070)	368.308 (2) *p* < 0.001
**Data description for cluster**	**1** **(*n* = 47)**	**2** **(*n* = 104)**	**3** **(*n* = 68)**	**Combined** **(*n* = 219)**	**F(df)** ***p***
Years of seniority at work mean (SD)	16.980 (11.301)	19.490 (10.570)	5.760 (4.860)	14.690 (11.134)	45.897 (2) *p* < 0.001
**Data description for cluster**	**1** **(*n* = 47)**	**2** **(*n* = 104)**	**3** **(*n* = 68)**	**Combined** **(*n* = 219)**	**Chi Squared (df)** ***p***
Limited time employment contract fr (fr%)	15 (31.9%)	0	65 (95.6%)		162.584 (2) *p* < 0.001
Indefinite time employment contract fr (fr%)	32 (68.1%)	104 (100%)	3 (4.4%)		

## Data Availability

The datasets for this study are available from corresponding author on reasonable request.

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
