# Peer review of "Teaching in Times of the COVID-19 Pandemic: A Pilot Study on Teachers’ Self-Esteem and Self-Efficacy in an Italian Sample"

_ijerph, 2021, doi:10.3390/ijerph18158211_

Round 1

Reviewer 1 Report

The theme dealt with in the paper is of great importance for the pandemic period, especially in educational contexts. however, it is necessary to update the bibliography which on some topics is rather dated (eg 1197, 2001, 2003 etc.). Discussions should be more focused on the results produced. Finally, both in the introduction and in the discussion there are no studies on the same themes conducted in Italy, for example:

Toto, G. A., & Limone, P. (2021). Motivation, Stress and Impact of Online Teaching on Italian Teachers during COVID-19. Computers10(6), 75.

Toto, G. A., & Limone, P. (2021). From Resistance to Digital Technologies in the Context of the Reaction to Distance Learning in the School Context during COVID-19. Education Sciences11(4), 163.

address the same issues, have the same sample and were carried out in the same national context, it would be interesting to triangulate the results of these 3 studies.

Additional comments:

Items to improve:

  • update literature
  • analyze articles on the same topic (pilot study ... but there are 2 others similar indicates)
  • rewrite better discussions  

The research question investigates the impact of the Covid-19 pandemic on teachers in association with the levels of self-esteem and sense of self-efficacy, the problems in modulating distance learning and the perceptions of students' difficulties.

The research question is relevant and interesting as the literature relating to the impact of COVID-19 on the educational institution is still scarce and under development. The theme is relevant for education and training and also has a broad-spectrum socio-cultural value.

The treated topic is current as it investigates the effects of distance learning, which still mediates training in all educational contexts. Investigations about the teachers' point of view are lacking in the literature, which is why the analysis of the effects of the pandemic on the school institution is currently partial. The innovation of the study is located in the chosen sample, as the subjects studied are not the students but the teachers, and in the object of the study, relating to psychological constructs and not to performance results. Knowing the phenomenon through the perceptions of all the players in the school world allows us to build a complete and detailed vision, and to modulate training and educational interventions that are effective in mediating a state of well-being and learning.

Studies relating to the school context, from 2020 to today, are centred on school inclusion in DDA, digital skills of teachers, technological tools that mediate effective learning, student perceptions and innovative teaching strategies. The psychological and working well-being of teachers during the pandemic is not tested. The research centred on teachers in the literature is oriented on the organisation and work planning, on the evaluation of students, and on the perceptions of trainees with respect to the use of technologies.

The research in question is therefore original as it investigates psychological constructs such as self-esteem, the sense of self-efficacy and the perception of teachers about the difficulties of their students during distance learning. Understanding the psychological dynamics of teachers provides useful information for improving their working well-being.

The text is clear and easy to read. The introduction illustrates the school context of distance learning with respect to where to place the research question; however, there are no references to similar research centred on the same theme. Each step of the study is well exposed: the description of the sample is precise, the explanation of the constructs is precise, the tools used to measure the constructs are well specified, each statistical procedure is well explained, and the tables allow a global and rapid understanding of the data.

The results that emerged, through statistical analysis, show a level of self-esteem and self-efficacy of teachers that is lower than that of the normative sample, and teachers also consider students' difficulties to be greater than those they themselves encounter. The problems that emerged relate to internet connection, an indispensable tool for accessing distance learning and relational dynamics. What emerged responds to the initial research question and is consistent with the conclusions that are drawn: attention to the organisational aspect of training favours cognitive goals, yet forgets the affective dynamics. This phenomenon represents an obstacle to student learning and complicates teaching for teachers. The authors also cite the limitation of the research inherent in the small number of participants. The association between relational variables and the learning process is not yet confirmed.

The centrality of the narrative derives from the studies of authors such as Bruner. Bruner is an American cultural psychologist who spent a significant part of his research in claiming an important role in psychological research for narrative thinking. Storytelling is important because everyone has a story to tell, and having the tools to effectively tell a story means having a voice and therefore the ability to express oneself and to be able to exercise a role of active citizenship in the digital universe.

Author Response

Dear Editor and Referees,

Thank you for your letter Ref. Manuscript ID: IJERPH -1312308-R1, entitled “Teaching in Times of COVID-19 Pandemic: a Pilot Study on Teachers' Self-esteem and Self-efficacy in an Italian Sample”, and for giving us the opportunity to review and resubmit the paper.

We are very grateful to your and the reviewers’ comments and suggestions; we are deeply appreciative of your careful reading.

Detailed replies to your comments are enumerated below, with the list of modifications and integrations. We hope this revised version now satisfies the requirements for publication in your journal.

Then, we submit the revised version of paper; for clarity new portions, added or modified in response to the referees’ comments, are highlighted in the manuscript; furthermore, the tracked version of the manuscript is attached.

Thank you very much

The Authors

Reviewer 2 Report

In general, the paper topic is timely and interesting. However, there are a number of errors in word usage and grammar that make some parts confusing. It would benefit from editing by a native English speaker.

In Line 13 (Abstract) and 96, the abbreviation LDs is used without explanation - from the context of the paper, I assume this is Learning Disabilities, but it should be defined in these two spots as the first usage.

Lines 211-213 and 299-302 appear to be left over from the template and should be removed from the manuscript.

The Discussion section is very short. Please elaborate on what the data results mean and how the lessons learned may be applied to classrooms (virtual/distance or face-to-face) in the future.

Also a question: did you look at your sample with the teachers separated by grade band? I would expect there may be different results for early grades teachers compared to teachers of older (age 15-18) students, for instance.

Author Response

(The authors gave the same response as above.)

Round 2

Reviewer 2 Report

Thank you for thoroughly addressing my concerns from the last review. The additional literature and discussion significantly improved the manuscript. There are still some language concerns and places where the abbreviation needs to be checked (e.g., LDs, Lds).